# Novel Impedance-pH Parameters in Pre-Bariatric Assessment of Patients: A Pilot Study

**DOI:** 10.3390/jcm12030940

**Published:** 2023-01-25

**Authors:** Mario Gagliardi, Antonella Santonicola, Rossella Palma, Luigi Angrisani, Nigel J. Trudgill, Paola Iovino

**Affiliations:** 1Gastroenterology Unit, Department of Medicine, Surgery and Dentistry “Scuola Medica Salernitana”, University of Salerno, 84081 Baronissi, Italy; 2Department of Surgical Sciences, “Sapienza” University of Rome, 00168 Rome, Italy; 3Department of Public Health, “Federico II” University of Naples, 80131 Naples, Italy; 4Department of Gastroenterology, Sandwell and West Birmingham Hospitals NHS Trust, Birmingham B18 7QH, UK

**Keywords:** bariatric surgery, obesity, GERD, MNBI, PSPW

## Abstract

Novel impedance-pH parameters, Mean Nocturnal Baseline Impedance (MNBI) and Post-Reflux Swallow-Induced Peristaltic Wave (PSPW) index, have been proposed to improve the gastro-esophageal reflux disease (GERD) diagnostic yield. This study aims to determine the integrity of the esophageal epithelial barrier and chemical clearance using these novel parameters and to correlate them with acid exposure time (AET) and acid clearance time (ACT) in obese patients who are candidates for bariatric surgery (BS). Twenty impedance-pHmetry tracings of patients prior to BS were reviewed. Nine (45%) patients with a conclusive diagnosis of GERD had significantly higher ACT, lower MNBI in the distal esophagus and lower PSPW indexes compared to obese patients without GERD. Moreover, 100% of obese patients with GERD had a pathological ACT compared to obese patients without GERD (*p* = 0.003). However, the percentage of pathological MNBI and PSPW index did not differ between obese patients with and without GERD. The PSPW index and MNBI of the distal channel significantly correlated with ACT and AET. Further studies are needed to assess the role of time-consuming novel parameters in the routine evaluation of morbidly obese patients candidates for BS. The value of acid clearance time is confirmed as a relevant impedance-pH parameter in these patients.

## 1. Introduction

Gastro-esophageal reflux disease (GERD) is very prevalent in obese patients that it is noteworthy are often paucisymptomatic [1,2]. According to the recent Lyon consensus [3], a conclusive GERD diagnosis is achieved in a minority of patients by upper GI endoscopy demonstrating erosive esophagitis grade C or D, while the most useful parameter is still considered the acid exposure time that can be measured by 24 h pHmetry, or impedance-pH. The latter also allows the detection of all reflux episodes independently from their acidity. Another relevant parameter in obese patients with GERD is esophageal acid clearance that can be assessed by the esophageal acid clearance time, which is calculated by dividing the total acid exposure duration by the number of reflux episodes [4]. The acid clearance time in healthy subjects depends on primary peristalsis to clear most of the refluxate from the esophagus and then salivary bicarbonate to neutralize residual acid. Impaired chemical clearance plays an important role in determining the severity of reflux-induced esophageal mucosal damage [5].

Recently, two novel impedance parameters, namely the Mean Nocturnal Baseline Impedance (MNBI) and Post-Reflux Swallow-Induced Peristaltic Wave (PSPW) index, have been proposed to evaluate esophageal mucosal integrity and chemical clearance respectively, increasing the diagnostic yield of impedance-pH monitoring [6,7,8]. It has been already shown that defective chemical clearance is associated with reflux esophagitis and with impaired mucosal integrity as evaluated by MNBI in NERD [7].

There are no data on the use of these novel parameters in obese patients apart from a study reporting a low panesophageal MNBI in obese patients with or without GERD, suggesting the presence of an impaired epithelial barrier in the entire esophagus [6].

The present study aims to determine the integrity of the esophageal epithelial barrier and esophageal chemical clearance using the novel impedance parameters MNBI and PSPW index in a series of impedance-pH tracings obtained from morbidly obese patients (Class II and III) who are candidates for bariatric surgery and to evaluate the correlation of PSPW index and MNBI with acid exposure time and acid clearance time.

## 2. Materials and Methods

A total of 20 impedance-pH tracings obtained from our prospectively maintained database of obese patients candidates for bariatric surgery were selected. In these tracings, the catheter was well tolerated by the patients, and all 24 h impedance-pH recordings were completed.

The inclusion criteria were white females and males between the ages of 18 and 75, presence of Class II obesity (BMI > 35) with comorbidities (hypertension, dyslipidemia, type II diabetes mellitus, and respiratory diseases), or Class III obesity (BMI > 40) candidates for bariatric surgery, and all studies were performed off antacid/antisecretory medications for at least 21 days.

Exclusion criteria were previous major abdominal surgery and if endoscopy and additional evaluation showed any of the following: Barrett’s esophagus (BE), achalasia, infectious esophagitis, erosive esophagitis, or eosinophilic esophagitis. Impedance-pH studies were always preceded by high-resolution esophageal manometry (HRM) for accurate location of the lower esophageal sphincter (LES). All 24 h esophageal impedance-pH tracings were analyzed and stored in an electronic database. A conclusive diagnosis of GERD was reached according to Lyon Consensus on the basis of acid exposure time > 6% [3].

Ambulatory 24 h esophageal impedance-pH was performed with a mobile recording device and a catheter with 6 impedance channels and one pH channel (Sandhill Scientific ComforTec Highlands Ranch, CO, USA), following a standard protocol [9]. The data of all gastroesophageal refluxes were analyzed using a semiautomated software system (BioView, Sandhill Scientific, Highlands Ranch, CO, USA) and then verified manually by MG and AS, as previously reported [5].

MNBI was manually recorded in the two most distal channels, located at 3 and 5 cm above LES, during the night-time recumbent period. Three 10-min periods (around 1.00 a.m., 2.00 a.m., and 3.00 a.m.) were selected, and the software computed the mean baseline for each period. Periods including swallows, refluxes and pH drops were avoided. Then, the mean of the three measurements was manually calculated to obtain the MNBI. The cut-off for normal values was defined as >2292 ohms in the most distal channel [8].

A PSPW was defined as a swallow occurring within 30 s of a reflux episode, provoking an antegrade 50% drop in impedance relative to the pre-swallow baseline originating in the most proximal impedance site, reaching all the distal impedance sites and followed by at least 50% return to the baseline in the distal impedance sites. For each impedance-pH tracing, the number of refluxes followed within 30 s by a PSPW was manually divided by the number of total refluxes to obtain the PSPW index. The cut-off for normal values was defined as >61% [8].

### Statistical Analysis

The data are expressed in frequencies and percentages for qualitative variables and median (IQR) for quantitative ones unless otherwise indicated. Significance was expressed at a *p* < 0.05 level.

When appropriate, a χ2 test for categorical data, Mann–Whitney analysis and Spearman correlation for continuous data were used.

The SPSS for Windows version 15.0 statistical package (SPSS Inc., Chicago, IL, USA) was used for statistical analysis.

## 3. Results

Table 1 shows the impedance-pH parameters of 20 (65% female) morbidly obese patients candidates for bariatric surgery. There were no differences in gender, age and BMI among the two groups. Obese patients with GERD (45%) had significantly higher acid exposure time, longer acid clearance time both in total and in the supine position, higher number of acid reflux episodes in total, upright and in the supine position, lower number of non-acid reflux episodes both in total and in the upright position compared to those without GERD (Table 1).

There were no differences between obese patients with or without GERD in MNBI except for the most distal channel (channel 6), where a significantly lower MNBI was detected in obese patients with GERD. The PSPW index was significantly lower in obese patients with GERD. Moreover, 100% of obese patients with GERD had a pathological acid clearance time (*p* = 0.003) compared to obese patients without GERD. However, there was no significant difference between obese patients with or without GERD in the percentage of pathological MNBI (<2292 ohm in the most distal channel) or in the percentage of a low PSPW index (<61%).

The PSPW index and MNBI of the distal channel significantly correlated with acid clearance time R = −0.56 and R = −0.45, respectively (*p* < 0.05 in both cases) (Figure 1) and acid exposure time (R = −0.50 and R = −0.45, respectively, *p* < 0.05 in both cases) (Figure 2).

## 4. Discussion

In this study, the impedance-pH tracings obtained from two groups of morbidly obese patients who were candidates for bariatric surgery with and without a conclusive GERD diagnosis were evaluated. Our results showed that MNBI in the most distal channel of the catheter and the PSPW index was significantly lower in the obese patients with GERD, while acid clearance time was significantly longer. The acid clearance time correlated with the PSPW index and the MNBI of the most distal channel.

While the pathophysiological mechanisms that link GERD and obesity have not been fully clarified, several studies have shown that there is a higher rate of GERD among the obese population [10,11,12]. Moreover, acid clearance time has been previously considered the most relevant impedance-pH parameters in these patients [13]. Insufficient esophageal clearance of fluids and acid could lead to a prolonged exposure of esophageal mucosa to acid and other peptic constituents of the gastric content that will ultimately produce mucosal damage [5].

Our findings support previous studies suggesting that the contribution of obesity to GERD depends not only on the lower esophageal sphincter function and the pressure challenge across the gastro-esophageal barrier [13,14]. In the last years, two novel impedance parameters, namely MNBI and PSPW index, have been proposed in order to achieve a better GERD diagnostic yield, especially in case of inconclusive pH-impedance findings [3].

In addition to the acid clearance time, the novel PSPW index has been proposed to evaluate the intactness of esophageal peristalsis and the effectiveness of salivary chemical clearance [7,8]. Moreover, the PSPW index appears to be a promising parameter in predicting response to PPI therapy when evaluated both off- or on-PPI to significantly distinguish hypersensitive esophagus from functional heartburn independently from SAP/SI positivity and to distinguish PPI-refractory NERD from functional heartburn [15,16]. To the best of our knowledge, there is a lack of studies regarding the use of MNBI and PSPW index in obese population candidate to BS apart from Blevins et al., who found a diffuse impairment of the esophageal epithelial barrier in obese patients, with a lower panesophageal MNBI independently from the presence of GERD, with a lower MNBI just detected in the distal esophagus of obese patients with a conclusive diagnosis of GERD [6].

Our data demonstrated a significantly lower PSPW index in morbidly obese patients with GERD; however, the percentage of patients with abnormal PSPW values did not differ between obese patients with or without GERD. Moreover, we showed a significantly lower MNBI in the distal esophagus in the group with GERD, although there was no difference in the percentage of abnormal MNBI (channel 6) between obese patients with or without GERD taking into account the cut-off value of 2292 ohms [17]. This result is consistent with the data reported by Blevins et al. [6]. However, given that the results of baseline impedance are influenced by obesity, the MNBI cut-off for GERD diagnosis may not be reliable in obese patients.

By 2030, one in two adults is projected to have obesity, showing the endemic trait of this condition [18]. Bariatric surgery is far more effective than nonsurgical interventions for the treatment of morbid obesity, since it accomplishes sustained weight loss, reduces obesity-related comorbidities and improves quality of life; therefore, the demand for this procedure is growing fast [19]. We strongly agree that endoscopy and reflux monitoring should be part of the pre-operative work-up prior to bariatric surgery [13,20,21], since typical reflux symptoms do not predict the presence of esophagitis and abnormal acid exposure [3] and given that GERD obese patients are often asymptomatic [1,2]. The presence of conclusive GERD diagnosis should influence surgeons to choose the type of bariatric procedure to offer to the patients. Laparoscopic Roux-en-Y gastric bypass (RYGB) is the second most performed bariatric procedure, and it is considered the procedure of choice to treat obese patients with GERD, although some authors described the recurrence or the new onset of GERD symptoms after RYGB. Similarly, MNBI and PSPW index results should be taken into consideration if they provide additional data of a diagnosis of conclusive GERD [22].

Nevertheless, given the lack of reliability of MNBI in distinguishing GERD in obese patients and the strong correlation between automated acid clearance time calculation and both MNBI and PSPW index, the applicability of these novel parameters in the routine pre-surgical assessment of morbidly obese patients might be questioned. Current analysis software do not include the automatic calculation for these novel parameters; hence, a careful time-consuming manual review of tracings is required, and variability between reviewers in the analysis of MNBI and PSPWs has been demonstrated [23]. Moreover, a high variability of PSPW index in healthy subjects has been demonstrated, depending on the monitoring system employed [24], as well as a lack of studies evaluating the normal threshold of these novel parameters [25]. In fact, previous studies concluded that the PSPW index cannot be used as an isolated metrics to define GERD [23].

However, some limitations are associated with this study. First and foremost, it was a retrospective analysis of a small cohort of patients from a single-center database. Given this, there may be potential selection bias, although patients were prospectively enrolled.

Moreover, because of the small sample sizes, we acknowledge the risk of overestimating with limited degrees of freedom. Secondly, the study did not include non-obese healthy volunteers. However, this pilot study aimed to determine the integrity of the esophageal epithelial barrier and chemical clearance using these novel parameters, and their correlation with standard parameters, in obese patients candidate to BS.

## 5. Conclusions

The pathophysiological link between an impaired epithelial barrier, an abnormal refluxate clearance and gastroesophageal reflux episodes in obese patients deserves further investigation. Currently, the calculations of MNBI and PSPW index values are not included in current analysis software, while manual analysis and calculations are required with a complicated process leading to prolonged report time, which can delay the therapeutic decision-making process for the patient, increasing costs for the healthcare system.

For this reason, it is probably better to limit the use of these time-consuming novel parameters as isolated metrics to define GERD in the pre-bariatric surgery routine evaluation of morbidly obese, since further studies are needed to assess their clinical usefulness.

## Figures and Tables

**Figure 1 jcm-12-00940-f001:**
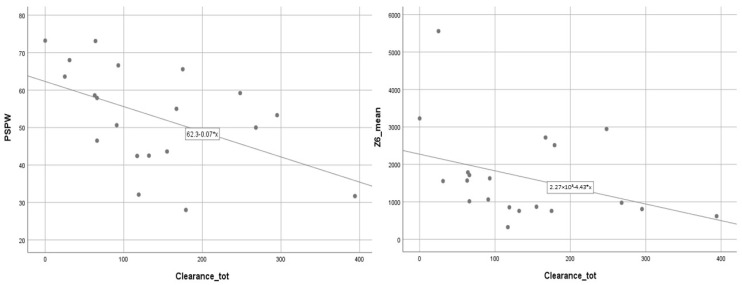
The correlation between PSPW index and MNBI of the distal channel and acid clearance time (*p* < 0.05 in both cases). Abbreviations: Mean Nocturnal Baseline Impedance (MNBI), Post-Reflux Swallow-Induced Peristaltic Wave (PSPW) index.

**Figure 2 jcm-12-00940-f002:**
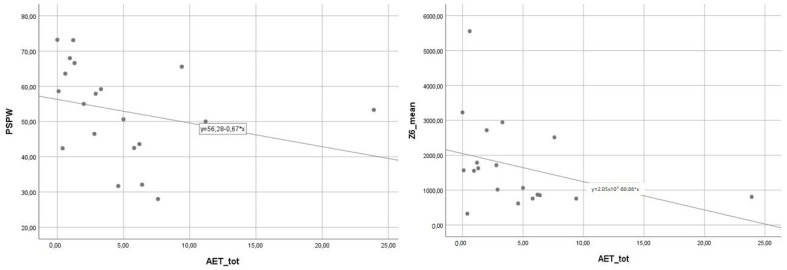
The correlation between PSPW index and MNBI of the distal channel and acid exposure time (*p* < 0.05 in both cases). Abbreviations: Mean Nocturnal Baseline Impedance (MNBI), Post-Reflux Swallow-Induced Peristaltic Wave (PSPW) index.

**Table 1 jcm-12-00940-t001:** Impedance-pH parameters of the studied obese patients candidate to bariatric surgery with and without proven GERD. The data are expressed in frequencies and percentages for qualitative variables and median (IQR) for quantitative ones unless otherwise indicated.

	n.9 (45%) OB with GERD	n.11 (55%) OB without GERD	*p*
Gender			
- Male	2 (22.2%)	5 (45.5%)	0.27
- Female	7 (77.8%)	6 (54.5%)	
Age	34.56 (27.6–41.45)	43.27 (36.59–49.96)	0.06
BMI (Kg/m^2^)	45.48 (42–48.95)	44.86 (41.63–48.09)	0.74
AET (%)	6.4 (5.4–10.3)	1.2 (0.4–2.8)	<0.001
DeMeester score	24.9 (18.95–29.6)	3.7 (2–8.9)	<0.001
Acid clearance time (s)			
- Total	175 (125.5–281.5)	66 (31–117)	0.003
- Upright	113 (55.5–247)	64 (31–117)	0.201
- Supine	368 (71.5–483)	0 (0–99)	0.007
OB Pts with ACT total > 74 s [4]	9 (100%)	4 (36.4%)	0.003
Number of total reflux episodes			
- Total	71 (55.5–80)	57 (29–71)	0.080
- Upright	59 (43.5–75.5)	39 (26–62)	0.152
- Supine	10 (3–19)	5 (1–13)	0.456
Number of acid reflux episodes			
- Total	47 (39–72.5)	16 (6–33)	<0.001
- Upright	39 (32–68.5)	14 (6–28)	0.001
- Supine	10 (2.5–17)	1 (0–4)	0.038
Number of non-acid reflux episodes			
- Total	8 (6.5–17.5)	20 (13–56)	0.020
- Upright	8 (4.5–15.5)	20 (11–39)	0.020
- Supine	1 (0–2.5)	4 (0–6)	0.261
MNBI (Ω)			
- Channel 1	3043.2	2553.6	0.152
	(2338.8–4678.3)	(1990.6–3049.5)	
- Channel 2	1976.5	1610.4	0.261
	(1773.35–2924.43)	(1167–2366.5)	
- Channel 3	1427.13	1739	0.824
	(951.95–3017.9)	(1041.8–2851.8)	
- Channel 4	2035.9	2321.4	0.456
	(1295.35–3073.75)	(1634.7–530.9)	
- Channel 5	1146.5	1493.7	0.80
	(837.05–1609.15)	(1346.1–2579.2)	
- Channel 6	855.4	1714.1	0.010
	(758.45–1019.68)	(1555.1–2944.9)	
OB Pts with MNBI < 2292 (Ω)	8 (89%)	7 (64%)	0.2
PSPW index (%)	43.6 (31.89–51.96)	59.2 (55–68)	0.007
OB Pts with PSPW index < 61%	8 (89%)	6 (55%)	0.1

## Data Availability

Data sharing not applicable.

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
