# Peer review of "Novel Impedance-pH Parameters in Pre-Bariatric Assessment of Patients: A Pilot Study"

_jcm, 2023, doi:10.3390/jcm12030940_

Round 1
Reviewer 1 Report
Gagliardi and colleagues conducted a cohort study to investigated relationship between acid exposure time (AET) and acid clearance time (ACT) and novel impedance-pH parameters, Mean Nocturnal Baseline Impedance (MNBI) and Post- 12 Reflux Swallow-Induced Peristaltic Wave (PSPW) index. I have some suggestion liseted below:
1. The type of surgery may influence the findings, such as sleeve gastrectomy versus gastric bypass. Therefore, the type of surgery may be listed.
2. The limitation of this study should be mentioned.
3. In the conclusion: It is probably better to limit the use of these time-consuming novel parameters as isolated metrics to define GERD in the pre-bariatric surgery routine evaluation of 193 morbidly obese since further studies are needed to assess their clinical usefulness. The difference of times consumed between different measured method may be compared to support the conclusion.
Author Response
Gagliardi and colleagues conducted a cohort study to investigated relationship between acid exposure time (AET) and acid clearance time (ACT) and novel impedance-pH parameters, Mean Nocturnal Baseline Impedance (MNBI) and Post- 12 Reflux Swallow-Induced Peristaltic Wave (PSPW) index. I have some suggestion listed below:
- The type of surgery may influence the findings, such as sleeve gastrectomy versus gastric bypass. Therefore, the type of surgery may be listed.
We thank the Reviewer for his/her careful reading. We strongly agree with the reviewer's comment regarding the influence of the type of bariatric surgery on the impedance-pH findings. Despite that, we conducted this pilot study during the pre-operative assessment of patients candidate to bariatric surgery, so all patients were naïve for metabolic/bariatric interventions. Anyway, following your inspiring comment, it would be interesting to conduct further studies evaluating the effect of the type of bariatric intervention on the impedance-pH findings.
- The limitation of this study should be mentioned.
We appreciated the Reviewer’s comment. We added the limitations of the study in the discussion section (page 6, lines 196-203) as Reviewer suggested.
- In the conclusion: It is probably better to limit the use of these time-consuming novel parameters as isolated metrics to define GERD in the pre-bariatric surgery routine evaluation of 193 morbidly obese since further studies are needed to assess their clinical usefulness. The difference of times consumed between different measured method may be compared to support the conclusion.
We thank the Reviewer for his/her precious comment. Currently, reporting times have not been reported in the literature, therefore we were unable to perform a time comparison. In our experience, we dedicated an average time of about 90 minutes for the calculation of MNBI and PSPW index. Anyway, this can be a topic of interest for further studies on these novel parameters.

Reviewer 2 Report
Many thanks for asking me to review the article "Novel impedance-pH parameters in pre-bariatric assessment of 2 patients: a pilot study" by Gagliardi M et al.
I think the article provides a novel concept for measuring the AET and ACT in obese patients preoperatively. However, the question remains, whether regular inclusion of these parameters (MNBI and PSPW) will influence the surgeon in choosing the type of bariatric procedure on an individual as some of the bariatric procedures are quite reflexogenic in nature. The second question is why not simply use the combination of HRIM and 24- or 48-hour pH study to identify these patients with GERD, which in my opinion is possible in almost 99% of patients. This is less time consuming and labor intensive. I also agree with the authors conclusions that "It is probably better to limit the use of these time-consuming novel parameters as isolated metrics to define GERD in the pre-bariatric surgery routine evaluation of morbidly obese since further studies are needed to assess their clinical usefulness" and I will also add the cost implications for both the patient and healthcare system.
Author Response
Many thanks for asking me to review the article "Novel impedance-pH parameters in pre-bariatric assessment of 2 patients: a pilot study" by Gagliardi M et al.
I think the article provides a novel concept for measuring the AET and ACT in obese patients preoperatively. However, the question remains, whether regular inclusion of these parameters (MNBI and PSPW) will influence the surgeon in choosing the type of bariatric procedure on an individual as some of the bariatric procedures are quite reflexogenic in nature.
We thank the Reviewer for his/her careful reading. We deeply appreciated the reviewer's suggestion, and we delved more closely into this concept in the discussion section (page. 6, lines 177-184)
The second question is why not simply use the combination of HRIM and 24- or 48-hour pH study to identify these patients with GERD, which in my opinion is possible in almost 99% of patients. This is less time consuming and labor intensive.
We thank the Reviewer for his/her comment. His/her observation agrees with the topic of this pilot study. Indeed, we aimed to evaluate if the time-consuming calculation of MNBI and PSPW index would provide additional information to the standard report of impedance-pH findings in the pre-bariatric assessment of obese patients. Moreover, due to the strict correlation between MNBI, PSPW index and ACT, we hypothesized that the rapid and automatic calculation of ACT can be an accurate surrogate for time-consuming calculations of MNBI and PSPW index.
I also agree with the authors conclusions that "It is probably better to limit the use of these time-consuming novel parameters as isolated metrics to define GERD in the pre-bariatric surgery routine evaluation of morbidly obese since further studies are needed to assess their clinical usefulness" and I will also add the cost implications for both the patient and healthcare system.
We thank the Reviewer for this comment. We deeply agree with his/her comment, so we modified the manuscript accordingly (page 6, lines 208-211).

Round 2
Reviewer 1 Report
I am grateful for the authors hard work. The questions were well addressed. Thank you very much.
Author Response
We thank the Reviewer for his/her precious comments.
